# INSTRUCTCV: INSTRUCTION-TUNED TEXT-TO-IMAGE DIFFUSION MODELS AS VISION GENERALISTS

**Yulu Gan**
Peking University

**Sungwoo Park**
UC Berkeley

**Alexander Schubert**
UC Berkeley and UCSF

**Anthony Philippakis**
Broad Institute of MIT & Harvard

**Ahmed M. Alaa**
UC Berkeley and UCSF

## ABSTRACT

Recent advances in generative diffusion models have enabled text-controlled synthesis of realistic and diverse images with impressive quality. Despite these remarkable advances, the application of text-to-image generative models in computer vision for standard visual recognition tasks remains limited. The current de facto approach for these tasks is to design model architectures and loss functions that are tailored to the task at hand. In this paper, we develop a *unified language interface* for computer vision tasks that abstracts away task-specific design choices and enables task execution by following natural language instructions. Our approach involves casting multiple computer vision tasks as text-to-image generation problems. Here, the *text* represents an *instruction* describing the task, and the resulting *image* is a visually-encoded *task output*. To train our model, we pool commonly-used computer vision datasets covering a range of tasks, including segmentation, object detection, depth estimation, and classification. We then use a large language model to paraphrase prompt templates that convey the specific tasks to be conducted on each image, and through this process, we create a multi-modal and multi-task training dataset comprising input and output images along with annotated instructions. Following the InstructPix2Pix architecture, we apply instruction-tuning to a text-to-image diffusion model using our constructed dataset, steering its functionality from a generative model to an instruction-guided multi-task vision learner. Experiments demonstrate that our model, dubbed *InstructCV*, performs competitively compared to other generalist and task-specific vision models. Moreover, it exhibits compelling generalization capabilities to unseen data, categories, and user instructions.

**Code:** https://github.com/AlaaLab/InstructCV

**Demo:** https://huggingface.co/spaces/alaa-lab/InstructCV

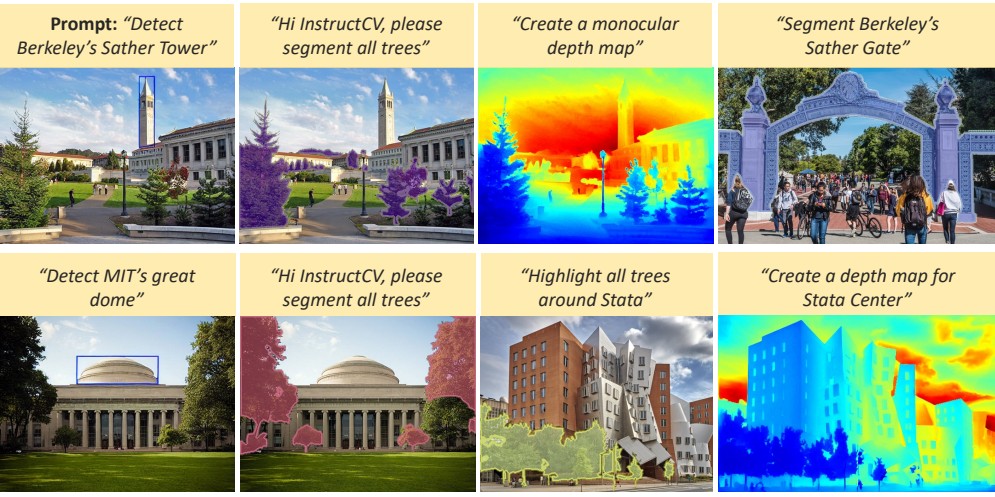

Figure 1: **Application of InstructCV to new test images & user-written instructions:** InstructCV performs the vision task described in the instruction on the input image. *(Images courtesy of UC Berkeley and MIT).*

# 1 INTRODUCTION

Recent work on text-to-image models has achieved impressive performance in image synthesis [1–3]. Particularly, diffusion models [4–7] have demonstrated remarkable capabilities of transforming diverse text prompts into realistic images, even for novel concepts. Models like DALL·E [2] and Stable Diffusion [8] highlight this progress, now finding use in real-world applications. However, despite these impressive results, generative text-to-image models have so far not been exploited as a unified basis for standard visual recognition tasks. Instead, the predominant approach for these tasks is to design dedicated task-specific architectures and loss functions [9–11], foregoing the opportunity to learn generalizable representations across heterogeneous problem domains and data landscapes.

Previous attempts to create unified models for computer vision tasks have predominantly relied on *prompt tuning* approaches in conjunction with sequence-to-sequence architectures [9–17]. This general framework enables conditioning on input images as well as task-specific prompts by representing image pixels and (trainable) prompts as sequences of discrete tokens. When trained on multi-task datasets, the resulting output tokens align with the desired outcomes for the respective tasks prompted. One illustrative example of this approach is Pix2Seq [16], which follows an autoregressive language modeling approach for processing tokenized image pixels and task identifier codes. Another example includes a class of methods based on *visual prompting*, which defines task-specific prompts in pixel space, unifying multiple vision tasks within a common "inpainting" framework [18, 19]. In both examples, the task-specific prompts steer a single architecture to execute multiple vision tasks. However, these prompts consist of (uninterpretable) numerical values derived from specific training datasets, which may limit their ability to generalize to new datasets, tasks, or categories.

In this paper, we propose a unified model for computer vision tasks that conducts a given task by following natural language instructions (Fig. 1). Our framework, dubbed *InstructCV*, repurposes generative text-to-image models to create a universal language interface for vision tasks. It does so by casting multiple computer vision tasks as text-to-image generation problems, where textual prompts (instructions) serve as explicit task descriptors, guiding the generation process to produce the visual task output corresponding to the input image. By conditioning on natural language descriptions of vision tasks, InstructCV enhances the representation of semantic coherence between images and language prompts, improving the model's generalization capabilities to new human-written instructions and new categories compared to prior "generalist" vision models [12, 13, 15–26].

To train InstructCV, we follow an *instruction tuning* approach applied to a pretrained conditional diffusion model (Stable Diffusion). We generate the instruction tuning data by constructing a multimodal, multi-task training dataset that comprises tuples of textual instructions, input images and visually-encoded task outputs. We do so by first combining several standard computer vision datasets across multiple tasks including segmentation, object detection, depth estimation, and classification. Next, in order to create heterogeneous and semantically rich textual instructions, we use a large language model (LLM) to paraphrase prompt templates for each vision task. Finally, we encode the output of the vision task associated with each instruction in the form of an output image (e.g., a masking pattern for semantic segmentation). Using this dataset, we utilize the InstructPix2Pix architecture [27] to instruction-tune a text-to-image diffusion model, transforming its functionality from a generative image synthesis model into an instruction-guided multi-task vision learner.

Our experiments demonstrate that InstructCV achieves competitive results compared to other vision generalist and task-specific vision models. Particularly, InstructCV displays compelling generalization properties, surpassing the performance of state-of-the-art vision generalist models on external datasets as well as on unseen prompts in open-vocabulary segmentation tasks.

# 2 INSTRUCTCV

The InstructCV framework comprises two key steps: (a) construction of a multi-modal and multi-task instruction-tuning dataset, and (b) finetuning a pretrained conditional diffusion model using the dataset generated in step (a). (See Fig. 2 for a pictorial depiction.) Details of each step are provided below.

## 2.1 CONSTRUCTING A MULTI-MODAL & MULTI-TASK INSTRUCTION-TUNING DATASET

We combine four widely-used computer vision datasets (MS-COCO [28], ADE20K [29, 30], Oxford-III-Pets [31] and NYUv2 [32]) covering four vision tasks (semantic segmentation, object detection,

Figure 2: **Pictorial depiction of the InstructCV training pipeline.** (a) We pool multiple computer vision datasets to construct a multi-modal and multi-task set of image pairs, where the target of each task is visually encoded in the form of an output image. Starting with a set of task-specific prompt templates, we sample a new instruction for each training point by using an LLM to rephrase the template for the corresponding task. (b) Using the dataset in (a), we finetune a diffusion model to produce the output $\mathbf{v}(\mathbf{y})$ given an image $\mathbf{x}$ & an instruction $\mathcal{I}$.

monocular depth estimation and classification), into a single multi-task dataset $\mathcal{D} = \{(\mathbf{x}_i, \mathbf{y}_i, m_i)\}_i$ in which $\mathbf{x}_i$ is the input image, $\mathbf{y}_i$ is the task output and $m_i \in \{1, \dots, M\}$ is the task identifier ($M = 4$ in our setup). We convert $\mathcal{D}$ into a multi-modal instruction-tuning dataset $\mathcal{D}_\mathcal{I} = \{(\mathbf{x}_i, \mathbf{v}(\mathbf{y}_i), \mathcal{I}_i)\}_i$, in which the task identifier for training point $i$ is expressed as a natural language instruction $\mathcal{I}_i$, and the label $\mathbf{y}_i$ is represented in a visual format $\mathbf{v}(\mathbf{y}_i)$. We construct the dataset $\mathcal{D}_\mathcal{I}$ through the following steps.

**LLM-based instruction generation.** For each vision task $m \in \{1, \dots, M\}$ under consideration, we pick a prompt template $\mathcal{I}_{\text{temp}}^m$ that describes the task, e.g., *"Segment the %category%"* for the semantic segmentation task. We then attach a prompt to each training data point by inserting the category within the image in the corresponding task template, e.g., $\mathcal{I}_{\text{temp}}^m(\mathbf{x}_i, \mathbf{y}_i) = $ *"Segment the cat"*. We consider two incarnations of our instruction-tuning dataset. First, we consider a baseline dataset that only uses the deterministic task-specific prompt templates described above. We refer to this dataset as the fixed prompts (FP) instruction-tuning dataset $\mathcal{D}_\mathcal{I}^{\text{FP}} = \{(\mathbf{x}_i, \mathbf{v}(\mathbf{y}_i), \mathcal{I}_{\text{temp}}^{m_i}(\mathbf{x}_i, \mathbf{y}_i))\}_i$. In addition, we use a T5-based paraphrasing LLM [33, 34] to generate rephrased versions of the prompt template to create a diverse range of instructions. As shown in Fig. 2(a) (top), the LLM takes as an input the prompt template (e.g., *"Segment the %category%"*) and produces a wide range of paraphrased variants (e.g., *"Highlight the %category%"*). This procedure ensures that our instruction set is varied yet firmly tied to the core intent of the original prompt. We use the LLM to sample a rephrased variant of the prompt template $\mathcal{I}_i \sim \text{LLM}(\mathcal{I}_{\text{temp}}^{m_i})$ for each training data point $i$ in $\mathcal{D}$. We refer to the resulting instruction tuning dataset as the rephrased prompt (RP) dataset $\mathcal{D}_\mathcal{I}^{\text{RP}} = \{(\mathbf{x}_i, \mathbf{v}(\mathbf{y}_i), \mathcal{I}_i(\mathbf{x}_i, \mathbf{y}_i))\}_i$.

**Visual encoding of task outputs.** We format the target label $\mathbf{y}$ of each task to represent it in the same RGB image space as the input image $\mathbf{x}$ through a "visual encoding" function $\mathbf{v}(\mathbf{y})$ (Fig. 2(a) (bottom)). This enables casting all tasks in a unified text-to-image generation framework, leveraging Pix2Pix architectures. That is, given an image $\mathbf{x}$ and instruction $\mathcal{I}$, InstructCV produces an image $\mathbf{v}(\mathbf{y})$ that encodes the task output. In the following, we provide the definition of $\mathbf{v}(\mathbf{y})$ for all tasks under study.

*(1) Semantic Segmentation.* The target output $\mathbf{y}$ of this task is typically an assignment of a label or category to every pixel in an image. A natural choice of $\mathbf{v}(\mathbf{y})$ for the semantic segmentation task is a binary mask that labels pixels in the input image $\mathbf{x}$ belonging to the prompted category in $\mathcal{I}$.

*(2) Object Detection.* Here, the goal is to identify the spatial position of a category in an image using a bounding box, i.e., the label comprises bounding box coordinates $\mathbf{y} = [c_x, c_y, w, h]$. We define $\mathbf{v}(\mathbf{y})$ for object detection as the image $\mathbf{x}$ with a bounding box overlaid according to the coordinates in $\mathbf{y}$.

*(3) Monocular Depth Estimation.* The target $\mathbf{y}$ of this task is the depth value (i.e., distance relative to the camera) of each pixel in the RGB image $\mathbf{x}$. For this task, we define the visually-encoded target

$\mathbf{v}(\mathbf{y})$ as an RGB image in which pixel colors encode the depth values. This encoding is done by converting depth values ranging from 0 to 10 meters (based on depth ranges in the NYUv2 dataset [32]) into the discrete space $[0, 1, \ldots, 255]$ for RGB image representation, i.e., $\mathbf{v}(\mathbf{y}) = \lfloor \mathbf{y} \times \frac{255}{10} \rfloor$. We then apply the same value across all three RGB channels to create a visual depth map.

***(4) Image Classification.*** In multi-class image classification, the target label $\mathbf{y}$ is a categorical value indicating the object depicted in the image $\mathbf{x}$. To represent image classification in a Pix2Pix format, we resort to a color-coding methodology. To this end, we use a prompt template of the form: *"Display %color% if the image contains %category%"*. We sample random colors when filling in the template for individual training points. This steers the text-to-image model to produce an image consisting of the pure color block if the category specified in the prompt is visible in $\mathbf{x}$. (Note that this approach only enables us to predict if a specific category is in the input image $\mathbf{x}$. For multi-class classification, we need to use a series of prompts specifying all categories of interest one at a time.)

## 2.2 INSTRUCTION-TUNING A LATENT DIFFUSION MODEL

We use our instruction-tuning dataset $\mathcal{D}_{\mathcal{I}} = \{(\mathbf{x}_i, \mathbf{v}(\mathbf{y}_i), \mathcal{I}_i)\}_i$ to train a (conditional) diffusion model that conducts the vision task specified in the instruction $\mathcal{I}$ on the input image $\mathbf{x}$, producing a visually-encoded task output $\mathbf{v}(\mathbf{y})$. By finetuning the text-to-image diffusion model using $\mathcal{D}_{\mathcal{I}}$, we steer its functionality from a generative model to a language-guided multi-task vision learner. We use a training procedure similar to that of InstructPix2Pix [27], which uses a similar multi-modal dataset (pairs of images and editing instructions) to train an instruction-guided image editing model.

Diffusion models [7] generate data by gradually denoising a normally distributed random variable; this process amounts to learning the reverse dynamics of a Markov chain with fixed length $T$. Latent diffusion models [8] apply this approach within the latent space of a pretrained variational autoencoder [35] with encoder $\boldsymbol{E}(.)$ and decoder $\boldsymbol{D}(.)$. Training a diffusion model involves a forward *diffusion* and a reverse *denoising* process. During the forward process, the image $\mathbf{v}(\mathbf{y})$ is transformed to its latent representation $\mathbf{z}_0 = \boldsymbol{E}(\mathbf{v}(\mathbf{y}))$, which is then injected with Gaussian noise over $T$ steps:

$$q(\mathbf{z}_t|\mathbf{z}_{t-1}) = \mathcal{N}(\mathbf{z}_t; \sqrt{1 - \beta_t}\mathbf{z}_{t-1}, \beta_t \mathbf{I}), \ \forall t \in \{1, \ldots, T\}, \tag{1}$$

where the time-varying constants $\beta_{1:T}$ control how much noise is added at each timestep $t$ and are chosen such that $z_T$ roughly converges to a standard Gaussian vector. This forward process does not contain any trainable parameters and can be described as $q(\mathbf{z}_{1:T}|\mathbf{z}_0) = \prod_{t=1}^{T} q(\mathbf{z}_t|\mathbf{z}_{t-1})$. In the reverse diffusion process, $p_\theta(\mathbf{z}_{0:T}) = p(\mathbf{z}_T) \prod_{t=1}^{T} p_\theta(\mathbf{z}_{t-1}|\mathbf{z}_t)$ the objective is to learn a model to progressively denoise the latents $\mathbf{z}_{T:1}$ to recover the initial latent $z_0$. The target image can then be reconstructed as $\mathbf{v}(\mathbf{y}) = \boldsymbol{D}(\mathbf{z}_0)$. The reverse diffusion process can be written as:

$$p_\theta(\mathbf{z}_{t-1}|\mathbf{z}_t) = \mathcal{N}(\mathbf{z}_{t-1}; \mu_\theta(\mathbf{z}_t, t), \rho_t^2 \mathbf{I}), \ \forall t \in \{1, \ldots, T\}, \tag{2}$$

where the means $\mu_\theta$ are typically parameterized using neural networks and the variances $\rho_t^2$ are predetermined constants. Such a denoising model is learned by optimizing a reweighted variant of the variational lower bound on the data distribution [4, 36], i.e.,

$$\mathcal{L}_{\text{unconditional}} := \mathbb{E}_{\boldsymbol{E}(\mathbf{v}(\mathbf{y})), \boldsymbol{\epsilon} \sim \mathcal{N}(0,1), t} \left[ \|\boldsymbol{\epsilon} - \boldsymbol{\epsilon}_\theta(t, \mathbf{z}_t)\|_2^2 \right], \tag{3}$$

where $\boldsymbol{\epsilon}$ is a standard normal random variable and $\mathbf{z}_t = \alpha_t \mathbf{z}_0 + \sigma_t \boldsymbol{\epsilon}$, where $\alpha_t^2 = \prod_{s=1}^{t}(1 - \beta_s)$ and $\sigma_t^2 = 1 - \alpha_t^2$ are based on the diffusion distribution $q(\mathbf{z}_t|\mathbf{z}_0) = \mathcal{N}(\mathbf{z}_t; \alpha_t \mathbf{z}_0, \sigma_t^2 \mathbf{I})$. The noise predictor $\boldsymbol{\epsilon}_\theta$ is obtained from the parameterization $\mu_\theta(\mathbf{z}_t, t) := (\mathbf{z}_t - \beta_t \boldsymbol{\epsilon}_\theta(\mathbf{z}_t, t)/\sqrt{1 - \alpha_t^2})/\sqrt{1 - \beta_t}$. The model $\boldsymbol{\epsilon}_\theta$ is trained to predict the noise vector $\epsilon$ at each time-step $t$ in order to denoise the latent variable $\mathbf{z}_t$.

**Instruction-tuning via image and text conditioning.** The objective in (3) can be further refined to condition on both the input image $\mathbf{x}$ and instruction $\mathcal{I}$ to generate the desired output $\mathbf{v}(\mathbf{y})$—this can be achieved by learning a conditional noise predictor [8, 27], minimizing the following loss function:

$$\mathcal{L}_{\text{conditional}} := \mathbb{E}_{\boldsymbol{E}(\mathbf{v}(\mathbf{y})), \boldsymbol{E}(\mathbf{x}), \mathcal{I}, \boldsymbol{\epsilon} \sim \mathcal{N}(0,1), t} \left[ \|\boldsymbol{\epsilon} - \boldsymbol{\epsilon}_\theta(t, \mathbf{z}_t, \boldsymbol{E}(\mathbf{x}), \mathcal{I})\|^2 \right]. \tag{4}$$

We use a pretrained Stable Diffusion checkpoint, which exhibits strong text-to-image generation capabilities, as the backbone architecture of our model. For text conditioning, we adopt the same methodology as in [8], utilizing the instruction $\mathcal{I}$ instead of image captions as textual inputs. For image conditioning, we concatenate the encoded input image $\mathbf{E}(\mathbf{x})$ with the latent $\mathbf{z}_t$, which are then fed into input to the first layer of the noise predictor $\epsilon_\theta$ [27].

**Classifier-free guidance.** To further improve the alignment between the generated outputs and their conditioning, we apply "classifier-free" guidance [37]. This approach combines the unconditional and conditional noise predictors in (4) and (3) in order to shift probability mass towards data where an implicit classifier $p_\theta(c|\mathbf{z}_t)$ assigns a high score to the conditioning $c$. Similar to InstructPix2Pix ((3) in [27]), we use a modified noise predictor that assigns different weights to the different components of the conditioning $(\mathbf{x}, \mathcal{I})$ as follows:

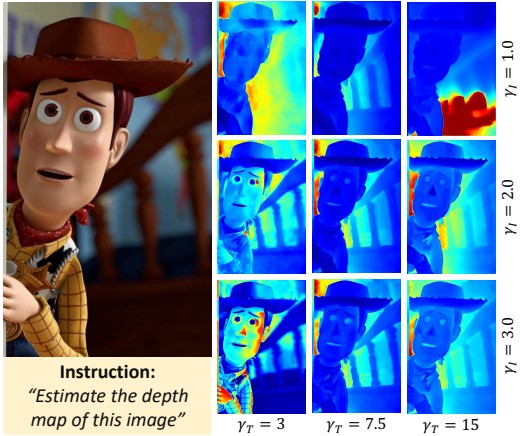

$$\begin{aligned}
\tilde{\epsilon}_\theta\left(t, \mathbf{z}_t, \mathbf{x}, \mathcal{I}\right) = {} & \epsilon_\theta(t, \mathbf{z}_t, \varnothing, \varnothing) \\
& + \gamma_I \cdot (\epsilon_\theta(t, \mathbf{z}_t, \mathbf{x}, \varnothing) - \epsilon_\theta(t, \mathbf{z}_t, \varnothing, \varnothing)) \\
& + \gamma_T \cdot (\epsilon_\theta(t, \mathbf{z}_t, \mathbf{x}, \mathcal{I}) - \epsilon_\theta(t, \mathbf{z}_t, \mathbf{x}, \varnothing)).
\end{aligned}$$

Figure 3: **Impact of classifier-free guidance** on the outputs of InstructCV for the depth estimation task.

where $\gamma_I$ and $\gamma_T$ are guidance scales that determine the relative importance of the image and text conditioning. Unconditional denoising is achieved by introducing null values to the respective image and text channels of the noise predictor. In Fig. 3, we show the effects of the parameters $\gamma_I$ and $\gamma_T$ on the output image $\mathbf{v}(\mathbf{y})$ of InstructCV. As we can see, $\gamma_I$ controls the similarity between the input and output images, $\mathbf{x}$ and $\mathbf{v}(\mathbf{y})$, whereas $\gamma_T$ controls the extent to which pixel values $\mathbf{v}(\mathbf{y})$ correspond to actual depth values in $\mathbf{x}$. Low $\gamma_T$ and high $\gamma_I$ result in output images that look very similar to input images with only a modification in the color map that does not reflect pixel depths. On the other hand, high $\gamma_T$ and low $\gamma_I$ produces coarse depth maps that miss nuanced details in the input images.

## 3 RELATED WORK

**Repurposing diffusion models for vision tasks.** Diffusion models have achieved impressive performance in image generation [4, 36, 38–40], text-to-image synthesis [1, 2, 6, 8], as well as generation of other modalities such as video [41] and audio [42]. The idea of repurposing diffusion models to tackle standard computer vision tasks has been considered before to develop models for object detection [43] and open-vocabulary panoptic segmentation [44]. These approaches were limited to single-task settings with specialized loss functions and (unimodal) architectures. Contrarily, InstructCV provides a unified architecture for multi-task learning, with a natural language interface that enhances generalization to new datasets and categories. The idea of "instruction-tuning" text-to-image diffusion models was introduced in [27] with the objective of finetuning the model to follow editing instructions. InstructCV builds on this framework to adapt text-to-image models for performing conventional visual recognition tasks. To our knowledge, this is one of the earliest efforts in this direction.

**Vision Generalists.** Several prior attempts have aimed to develop unified models capable of executing multiple vision tasks within a single, shared architecture. Motivated by successes of LLMs, recent work has attempted to design such generalist models based on sequence-to-sequence architectures. Among these, models such as Florence [9], OFA [10], CoCa [11] and BEiT-3 [14], learn general representation encoder, which require individual finetuning to each specific downstream task. Methods such as Unified-IO [15] and Pix2Seq-v2 [17] build single architectures that are capable of performing multiple vision tasks via prompt tuning. However, the sequence-based operation of these models results in slow inference speeds, and the tuned prompts may not generalize to unseen datasets/categories. Another line of work proposes vision transformer-based architectures that frame different vision tasks as inpainting problems [18, 19]. This work focuses on in-context learning based on visual prompts and does not consider language-based instructions, which we believe is a more natural interface for general-purpose models. To the best of our knowledge, the only generalist model that supports a language-based interface for vision tasks similar to that of InstructCV is the VisionLLM model developed in [13]. This is an LLM-based framework that treats images as a foreign language and aligns vision-centric tasks with language tasks that can be flexibly defined using language-based instructions. VisionLLM and InstructCV share a common objective but use different approaches. VisionLLM finetunes a pre-trained LLM using vision-centric tasks, whereas InstructCV finetunes a text-to-image model by substituting image captions with instructional text. We were unable to empirically compare the two models as the code for VisionLLM was not available at the time of writing this paper.

## 4 EXPERIMENTS

### 4.1 EXPERIMENTAL SET UP

We evaluate InstructCV across the four vision tasks under study (semantic segmentation, object detection, monocular depth estimation and image classification). For this purpose, we consider widely used datasets for each task: ADE20k [29, 30] for semantic segmentation, MS-COCO [28] for object detection, NYUv2 for depth estimation [45] and Oxford-IIIT Pet [46] for classification. In what follows, we explain the processing steps and evaluation procedure for all tasks under consideration.

**Semantic Segmentation.** ADE20K covers $150$ semantic categories and comprises $25,000$ images of which we use $20,000$ for training, $2,000$ for validation, and $3,000$ for testing. We follow the same protocol as suggested in [18] to implement the training/test split. At inference time, we average the outputs of the three channels of the output image $\mathbf{v}(\mathbf{y})$ to obtain the final segmentation mask. We evaluate the accuracy of segmentation masks using the Mean Intersection over Union (mIoU) metric.

**Object Detection**. MS-COCO contains $118,000$ training and $5,000$ validation images with labels for $80$ different categories. We follow the same protocol as in Pix2Seq [16] to set up the training/test split. At inference time, we follow the post-processing steps in Appendix A.3 to derive the coordinates and category of each Region of Interest (RoI) from the output image $\mathbf{v}(\mathbf{y})$. We then aggregate the results for all categories in order to calculate the Mean Average Precision (mAP).

**Depth Estimation.** The NYUv2 dataset [32] consists of $464$ indoor scenes captured by a Microsoft Kinect camera. We follow the official training/test split, with $24,231$ image-depth pairs used for training, and $654$ used for testing. For the test images we report the Root Mean Square Error (RMSE), absolute mean relative error (A.Rel), and the share of interior pixels with a different threshold $\delta$. During inference, we take the average across the three channels of the output image and apply the inverse of the linear transformation used in training to obtain a depth estimate in the range of $[0, 10]$ meters.

**Image Classification.** As mentioned in Section 2.1, we implement image classification by asking the InstructCV model if a category is visible in the input image $\mathbf{x}$ using the following template prompt: *"Display %color_1% if the image contains %category%, else display %color_2%"*. We evaluate the accuracy of InstructCV for classification by assessing whether $\mathbf{v}(\mathbf{y})$ contains the color block corresponding to the correct category. We do so by generating image pairs based on the Oxford-III Pet dataset for binary classification and augment these with negative pairs, where the category mentioned in the language instruction is not present. We then evaluate a classification score defined as: Cls-Score $(\mathbf{v}(\mathbf{y}), \mathbf{c}) = \sum_{i=1}^{n} \sum_{j=1}^{m} |\mathbf{v}_{i,j}(\mathbf{y}) - \mathbf{c}_{i,j}|$, i.e., the Euclidean distance between the pixel-wise colors of the output image $\mathbf{v}(\mathbf{y})$ and target color block $\mathbf{c}$ specified in the task instruction $\mathcal{I}$.

Our pooled multi-modal/multi-task instruction-tuning dataset comprises 180,285 images. We create two versions of the dataset, $\mathcal{D}_{\mathcal{I}}^{\mathrm{FP}}$ & $\mathcal{D}_{\mathcal{I}}^{\mathrm{RP}}$, with fixed and rephrased prompts as described in Section 2.1.

**External Datasets.** Since the generalist baselines and InstructCV were trained on different datasets[1], we consider additional external datasets that are outside of the training distribution of all baselines. To this end, we consider the following datasets: ImageNet [47] for classification, SUNRGB-D [48] for object detection and VOC [49] for segmentation and monocular depth estimation tasks.

**Implementation Details.** We train InstructCV for 20 epochs on 8 NVIDIA A100 GPUs over 10 hours. The training involves images with a resolution of $256 \times 256$ and incorporates data augmentation including random horizontal flipping and cropping with a batch size of $128$. The proposed model is initialized with EMA weights obtained from the Stable Diffusion checkpoint, and trained with a learning rate $10^{-4}$ without any warm-up stage. Further details can be found in appendix A.1. We refer to the models trained on $\mathcal{D}_{\mathcal{I}}^{\mathrm{FP}}$ and $\mathcal{D}_{\mathcal{I}}^{\mathrm{RP}}$ as InstructCV-FP and InstructCV-RP, respectively.

### 4.2 RESULTS

Table 1 presents a quantitative comparison between InstructCV and task-specific as well as generalist vision models in both in-distribution and out-of-distribution datasets. (Fig. 4 displays illustrative examples of InstructCV outputs.) For depth estimation, we compare InstructCV with DepthFormer [50], BinsFormer [51] and UviM [12]. For semantic segmentation, our baselines include Mask2Former [53]

---

[1]Unified-IO has been trained on 90 datasets while Pix2SeqV2 was trained on MS-COCO.

Table 1: **Comparison of InstructCV to task-specific and vision generalist baselines**. We report performance on segmentation, object detection, depth estimation and classification using test samples from the InstructCV instruction tuning dataset as well as external datasets for each task. InstructCV was evaluated using LLM-rephrased instructions for test data. (**Bold** and blue indicate best and second best performing model, respectively.)

| | Depth Estimation | | Semantic Segmentation | | Classification | | Object Detection | |
| | RMSE↓ | | mIoU↑ | | Acc↑ | | mAP@0.5↑ | |
| | NYUv2 | †SUNRGB-D | ADE-20K | †VOC | Oxford-Pets | †ImageNet-sub | COCO | †VOC |
|---|---|---|---|---|---|---|---|---|
| **Task-specific models** | | | | | | | | |
| DepthFormer [50] | 0.339 | 0.427 | | | | | | |
| BinsFormer [51] | 0.330 | 0.433 | | | | | | |
| SSA [52] | | | 47.150 | * | | | | |
| Mask2Former [53] | | | **56.100** | * | | | | |
| ResNet [54] | | | | | 97.500 | * | | |
| ViT [55] | | | | | **98.400** | * | | |
| DETR [28] | | | | | | | **60.600** | * |
| Mask R-CNN [56] | | | | | | | 60.500 | * |
| **Generalist framework, Task-specific models** | | | | | | | | |
| UviM [12] | 0.468 | * | | | | | | |
| **Generalist models** | | | | | | | | |
| Unified-IO [15] | 0.387 | 0.287 | 25.713 | 27.724 | 96.514 | **89.877** | | |
| Pix2SeqV2 [17] | | | | | | | 57.400 | 38.500 |
| Painter [18] | 0.288 | 0.285 | 49.9 | 52.5 | | | | |
| **InstructCV-FP** | **0.275** | **0.268** | 52.3 | **53.5** | 80.4 | 75.1 | 49.1 | **62.00** |
| **InstructCV-RP** | 0.297 | 0.279 | 47.235 | 52.125 | 82.135 | 74.665 | 48.500 | 61.700 |

† **External datasets**: These dataset were not part of the training dataset for the baselines under consideration.
* Task-specific models are incapable of directly implementing the corresponding task on external datasets if these datasets contain object categories that were not present in the models' training data.

and SSA [52]. For classification, we consider baseline classifiers with ResNet [54] and ViT [55] backbones. Lastly, for object detection, we consider DETR [28] and Mask R-CNN [56]. Task-specific models for semantic segmentation, classification, and object detection have not been assessed on datasets beyond their training distribution. This is due to the fact that these new datasets introduce categories absent in the model's original training, which precludes zero-shot generalization. We consider Unified-IO [15] and Pix2SeqV2 [17] as generalist vision baselines. Note that we only report object detection results for Pix2SeqV2 [17], as this model does not cover the other tasks involved in the development of InstructCV. Additional tasks Pix2SeqV2 is able to perform, such as keypoint detection, have not been integrated into the current version of InstructCV.

**Performance comparisons.** Overall, Instruct CV performs competitively compared to both generalist and task-specific baselines across all four tasks. For depth estimation within in-distribution data, InstructCV achieves a 10% improvement in RMSE compared to the second best model, BinsFormer [51]. Notably, InstructCV demonstrates strong generalization performance to unseen datasets, surpassing all baselines by a large margin, with the exception of classification tasks. For instance, for the task of depth estimation the task-specific

Table 2: **Open-vocabulary segmentation on the FSS-1000 dataset.**

| Model | mIoU ↑ |
|---|---|
| Inpainting [57] | 58.5 |
| Generalist Painter [18] | 62.3 |
| **InstructCV-RP** | **69.8** |

models Binsformer [51] and DepthFormer [50] experience high performance drops of $\approx 31.2\%$ and $\approx 26.0\%$, respectively, while InstructCV's performance improved by $6\%$ resulting in a $34.7\%$ lower RMSE than the best task-specific model. Similarly, for the task of object detection, the performance of the Pix2SeqV2 generalist model drops by $32.2\%$ when evaluated on VOC—InstructCV outperforms this generalist model by a $+23.2$ in mAP@0.5. Among all baselines, only Unified-IO demonstrates comparable generalization properties to unseen datasets. However, InstructCV outperforms Unified-IO on all tasks except for classification. Notably, for semantic segmentation, InstructCV surpasses the performance of Unified-IO by $+24.401$ in mIOU. Classification is the task where InstructCV exhibited its weakest performance compared to baselines.

**Generalization to unseen categories.** Most existing task-specific and multi-task models are built based on a fixed category pool determined by their training data, and do not exhibit zero-shot capabilities in detecting, segmenting or classifying categories outside of this set [15, 17, 28, 52–

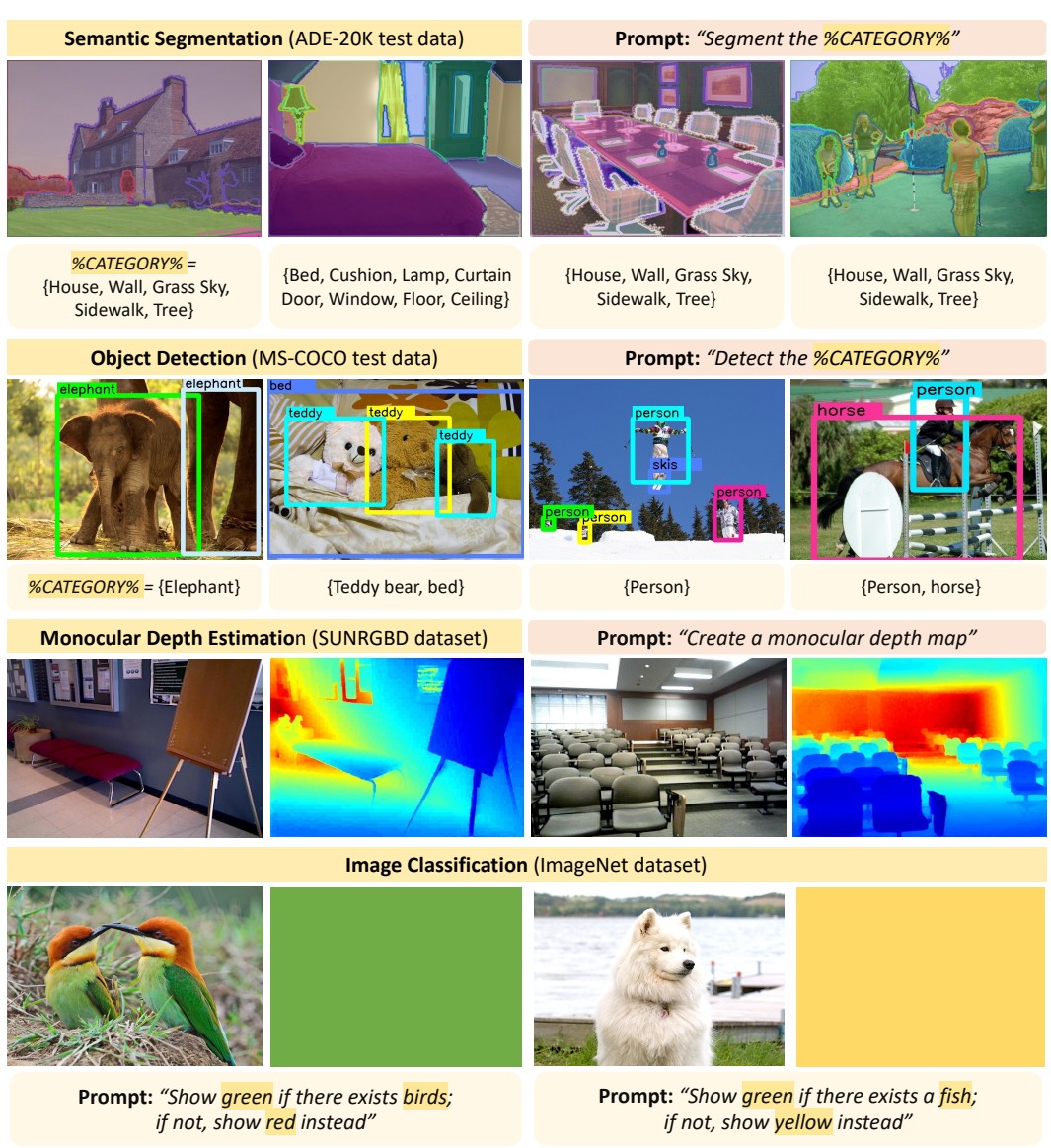

Figure 4: **Samples of InstructCV outputs across all vision tasks.** The segmentation and detection outputs (top two rows) are obtained by applying one prompt for each category and overlaying the results in one output image.

56]. Because InstructCV leverages semantically meaningful instructions and a pre-trained text-to-image generative model to guide learning, we expect its task-specific capabilities to generalize to new categories. To investigate its generalization capabilities to unseen categories, we evaluate InstructCV on an open-vocabulary segmentation task using the FSS-1000 dataset [58]. This dataset comprises 1000 object classes, many of which have not been previously annotated in other computer vision datasets. Because the baselines in Table 1 do not accommodate many of these categories, we instead compare InstructCV with generalist vision models that exhibit zero-shot capabilities—the visual prompting by Inpainting model in [57] and the Generalist Painter model in [18]. Both methods rely on visual prompting approaches, where the input prompt is a set of pixels and the vision tasks are all represented within a unified inpainting framework. Table 2 presents the results. Overall, InstructCV outperforms Generalist Painter and Inpainting by +3.5 and +7.3 in mIoU, respectively. This demonstrates the value of repurposing text-to-image models and language-based prompts to improve the generalization capabilities of generalist approaches to computer vision tasks.

**Generalization to new user-written instructions.** The InstructCV-RP model undergoes training using the dataset $\mathcal{D}_{\mathcal{I}}^{\text{RP}}$, which comprises a variety of instructions, all aimed at conveying a common

Table 3: **Evaluating InstructCV on new user-written instructions.** We evaluate the InstructCV-FP and InstructCV-RP models on the semantic segmentation task using test samples within the ADE20k dataset. At inference time, we apply new user-written instructions (listed below) that have not been used during training in $\mathcal{D}_{\mathcal{I}}^{\text{FP}}$ and $\mathcal{D}_{\mathcal{I}}^{\text{RP}}$, in addition to the template prompt (first row, shaded).

| Instruction | mIOU Score | |
| :--- | :---: | :---: |
| | **InstructCV-FP** | **InstructCV-RP** |
| **Template prompt:** *"Segment the %category%."* | 45.503 | 42.793 |
| *"Segment %category%."* | 36.315 | 48.739 |
| *"Can you help me segment the %category%?"* | 44.035 | 45.686 |
| *"Highlight the image parts corresponding to the %category%."* | 19.988 | 44.758 |
| *"Please highlight the image segment containing %category%."* | 21.952 | 40.153 |
| *"Which image segments contain the %category%?"* | 39.348 | 47.972 |

underlying intent (i.e., describing a specific visual task). Through this diverse training data that encompasses a broad spectrum of phrasings and descriptions of the same task, we expect that InstructCV-RP will be able to extrapolate its learning to novel user-generated instructions at inference time. To test this, we compared the performance of the InstructCV-FP and Instruct-RP variants on the semantic segmentation task. Both models were tested using manually-selected prompts (unseen in training data) on 200 images in the ADE20k test data. The results in Table 3 indicate that the InstructCV-RP model exhibits consistent performance and more robustness to variations in the phrasing of user instructions compared to InstructCV-FP. For example, testing InstructCV-FP with the instruction *"Please highlight the image segment containing %category%."* instead of the template prompt *"Segment %category%."* led to a $51.8\%$ drop in mIOU. Conversely, InstructCV-RP only incurred a performance reduction of $6\%$ with this instruction compared to the template prompt. This suggests that our LLM-based prompt rephrasing approach effectively enhances the ability of InstructCV to generalize to new user-generated prompts that convey descriptions of tasks similar to those seen during training.

**Computational costs.** Finally, we note that InstructCV was trained in an end-to-end fashion, with only 2,000 finetuning steps. The inference time of InstructCV on a single NVIDIA A100 GPU is 5 seconds (for a 256x256 image). This is a significant improvement over comparable generalist models, such as Unified-IO [15], which was trained from scratch using 1.5 million steps and takes around 40 seconds for inference on a single NVIDIA A100 GPU. Notably, InstructCV not only simplifies the training process but also outperforms Unified-IO in various tasks. These improvements can be attributed to the already impressive capabilities of the underlying text-to-image model. By instruction-tuning a generative model with a relatively small number of steps and a moderately-sized dataset, we are able to steer its functionality with performance that is competitive with bespoke generalist models.

## 5   CONCLUSION

In this paper, we introduce a unified language interface for computer vision tasks, dubbed InstructCV, eliminating the need for task-specific design choices and allowing for task execution based on natural language instructions. InstructCV frames various computer vision tasks as text-to-image generation problems. In this setup, textual instructions describe the task, and the resulting image serves as a visual representation of the task output. Following the InstructPix2Pix architecture, we curate a multi-task and multi-modal dataset to instruction-tune a pre-trained text-to-image diffusion model, steering its function from a generative model to an instruction-guided multi-task vision learner. By harnessing semantically meaningful language instructions to drive the learning process, our model demonstrates compelling generalization capabilities across unseen data, categories, and user instructions.

**Limitations.** The inference speed of our model lags behind specialized task-specific models and falls short of meeting the real-time inference requirements for tasks such as object detection and segmentation. Additionally, the semantic flexibility of InstructCV is constrained by the richness and diversity of our instruction-tuning dataset, which is currently generated by rephrasing a limited set of template prompts. This raises questions for future work: can this learning paradigm accommodate instructions that introduce more nuanced conditions? For example, an instruction might cap the count of objects to be detected. Exploring such ideas might require the integration of strategies such as learning from human feedback, which could enable more versatile generalist models by improving alignment of task outputs with more complex prompts.

## 6 ACKNOWLEDGEMENTS

The authors would like to thank David Sontag (MIT) for insightful feedback and discussions.

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

# A  APPENDIX

## A.1  INSTRUCTCV TRAINING DETAILS

We train our multi-task vision model across 20 epochs for 10 hours on an array of 8 80GB NVIDIA A100 GPUs. Our training utilizes images of $256 \times 256$ resolution and a batch size of 128. Augmentation techniques applied include random horizontal flipping and crop augmentation. For the latter, images are first subjected to random resizing between 256 and 288 pixels before being cropped to a 256-pixel size. We set our learning rate at 10e-4 without incorporating a learning rate warm-up phase. Model initialization is performed using the EMA weights from the Stable Diffusion v1.5 checkpoint, and we adopt other training settings from the public Stable Diffusion repository. For the results presented in this paper, we operate at a 256-pixel resolution with 100 denoising steps. We employ an Euler ancestral sampler with a denoising variance schedule as proposed by [59]. On an NVIDIA A100 GPU, our model takes approximately 10 seconds to solve a given vision task.

## A.2  EXAMPLES OF THE REPHRASED PROMPTS.

Figure 5-(a)/(b) qualitatively compares the robustness of InstructCV to changes in instruction wording when trained using the fixed prompt or the more diverse rephrased prompt dataset. The model trained using the fixed prompt data shows reasonable performance on segmentation tasks, even for prompts that slightly deviate from the standard instruction wording. However, as these deviations increase misclassfications become more common. Instead, the model trained on the rephrased prompt data appears more robust to such changes in task formulation. Notably, the model appears to show basic semantic understanding as it is in the last prompt example able to infer the correct intent despite the simultaneous occurrence of the potential object detection targets 'spider man' and 'face'.

## A.3  POST-PROCESSING STEPS FOR OBJECT DETECTION TASKS

We employ image processing techniques to derive the bounding box coordinates from the output image. First, we apply median and bilateral filters to the image in order to mitigate noise and enhancing features. Following, we convert the image from RGB to HSV space to isolate the red region within the target object, which is then extracted from the original image. Next, we identify closed contours in the image by converting it to a grayscale map, performing threshold segmentation based on grayscale, and ultimately, binarizing the image. However, naively applying this approach could potentially result in the removal of some accurate bounding boxes due to the disruptions induced by complex image backgrounds. To circumvent this issue, we cross-reference the bounding boxes with the dataset annotations and retain any bounding box predicted by the model that exhibits an Intersection over Union (IOU) greater than 0.5. We exclude disturbances that are not rectangular or that contain numerous red dots within the contour. The coordinates of the remaining contours are subsequently added to the prediction list.

Table 4: **Rephrased prompts**. We automatically generate diverse rephrased prompts to generated a diverse rephrased prompts (RP) instruction dataset. Here we display some examples of rephrased task templates.

| | Segmentation | Detection | Classification | Depth estimation |
|---|---|---|---|---|
| | Segment the %. | Detect the %. | Show * if there exists a % in the figure. | Estimate the depth of this image. |
| | Please break down the % into individual parts. | Can you help me detect the %? | Display an * if a % is present in the figure. | Approximate the depth of this image. |
| | Can you provide me with a segment of the %? | Please employ bounding boxes for the purpose of % detection. | In case a % exists in the figure, display an *. | Make an estimation of how deep the this image is. |
| Prompts | Please divide the % into smaller parts. | Locate the %'s presence. | If a % is present in the figure, indicate it with *. | Provide a rough calculation of the image's depth. |
| | Please perform image segmentation to isolate the % in this image. | Please use bounding boxes to identify the presence of a %. | If there is a % in the figure, show it as *. | Give an approximate measurement of the image's depth. |
| | Help me segment the %. | Detect and identify the %'s location. | Show * if the figure contains a %. | Make an informed guess of the depth of the image. |
| | Would you be willing to segment the %? | Utilize bounding boxes in order to identify the presence of the %. | Show an * if there is a % within the figure. | Make an estimation of how deep the image goes. |

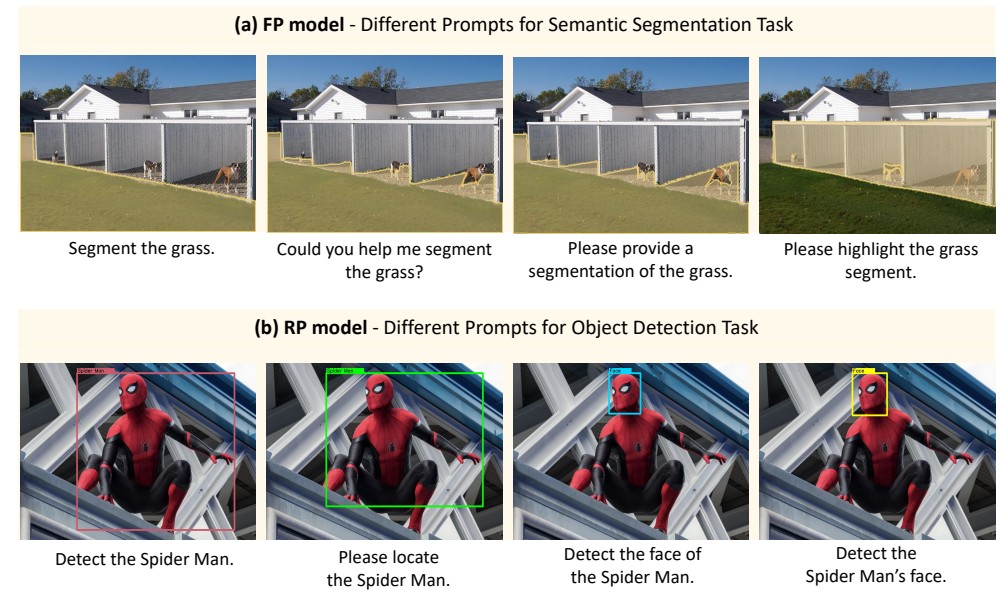

Figure 5: **Semantic segmentation performance for different prompt formulations** (a) **FP model**. Results for model trained on fixed prompt (FP) instruction dataset (b) **RP model**. Results for model trained on diverse rephrased prompt (RP) instruction dataset. Overall the RP model shows greater robustness to variations in wording.

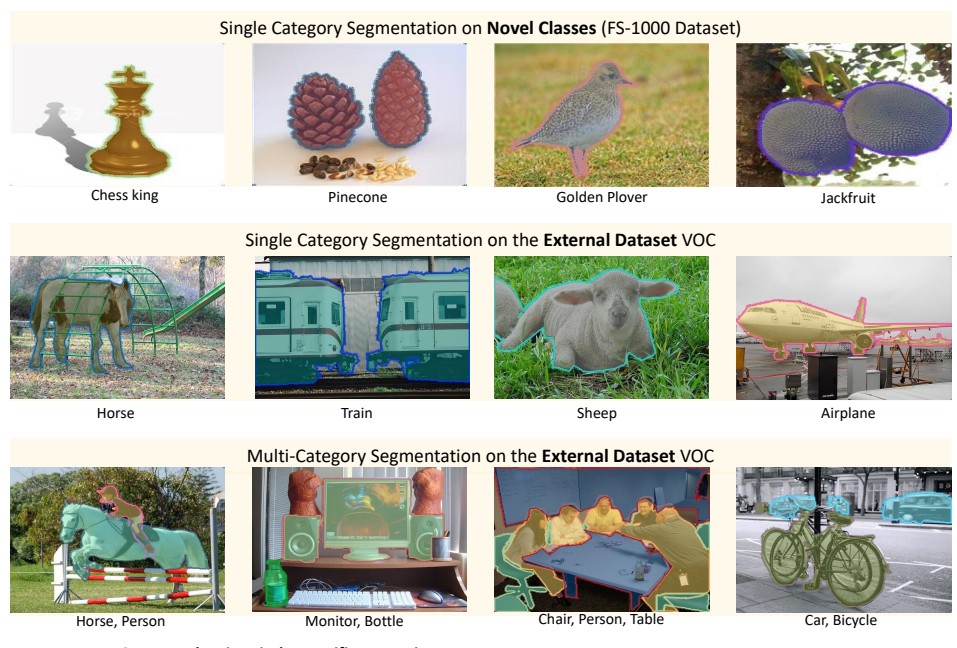

Figure 6: **Visualization of semantic segmentation examples.** Segmentation categories are provided below each image.

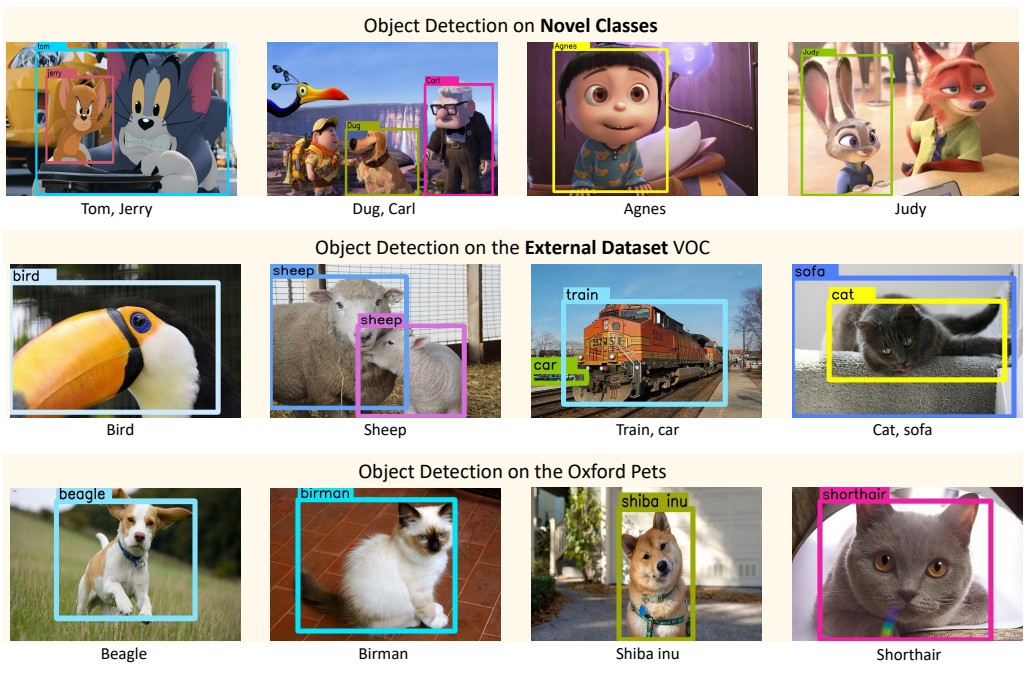

Figure 7: **Visualization of object detection examples.** Segmentation categories provided below the image.

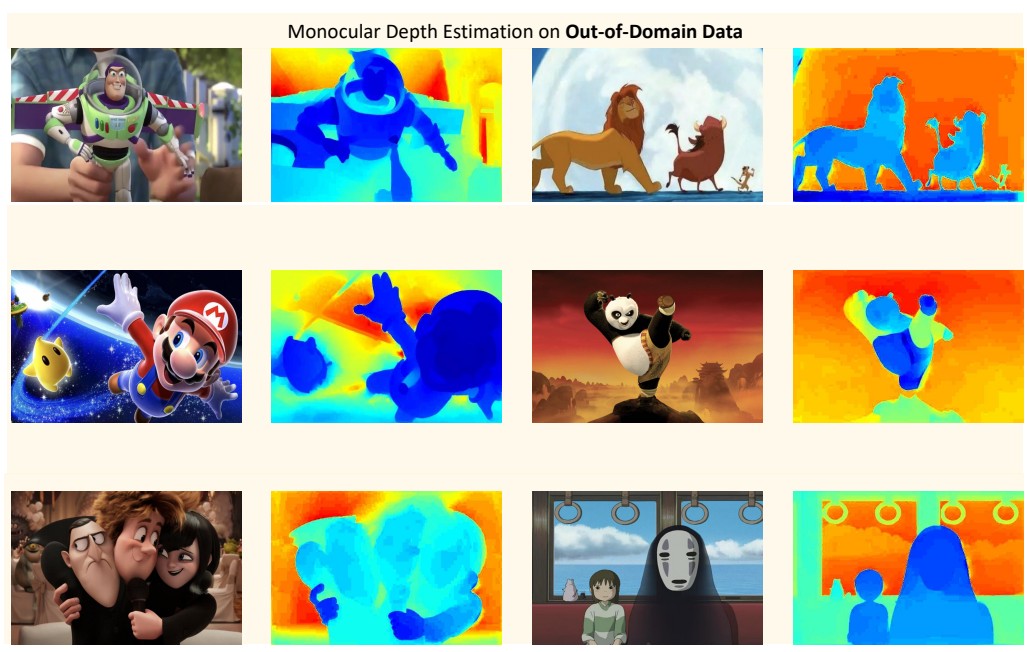

Figure 8: **Visualization of monocular depth estimation examples.** All depth maps presented here are produced for datasets not included during model training.

