# OpenReview forum: "InstructCV: Instruction-Tuned Text-to-Image Diffusion Models as Vision Generalists"
_ICLR.cc/2024/Conference — ICLR 2024 poster_

### Official Review · Reviewer_cJRy · 2023-10-29

**Soundness:** 2 fair
**Presentation:** 2 fair
**Contribution:** 2 fair
**Rating:** 3
**Confidence:** 5

**Summary:**

This paper slightly modifies the pipeline of InstructPix2Pix to use the diffusion model to achieve several compute vision tasks including segmentation, object detection, etc. The experimental results show that the proposed framework can be used to multiple tasks, but the performance is not very impressive.

**Strengths:**

1. This paper first uses instruct-tuning to adapt a pre-trained text-to-image diffusion model to several computer vision tasks.
2. This paper is well-written and easy to follow.

**Weaknesses:**

1. The novelty is limited. Formulating computer vision tasks as generative tasks is not novel and has been used in previous works[18]. The overall pipeline highly resembles InstructPix2Pix
2. The performance is not impressive at all. The results lag far from the SOTA methods in Table 1. Note that SOTA methods of depth estimation are not listed for comparisons and other bold results (VOC) are computed on very old benchmarks. The authors might argue that their method is not task-specific, but I think this is not the reason for such poor results.
3. The inference pipeline of image classification (Figure 4) seems really weird. I think it would be really hard to perform evaluation on 1000-class ImageNet.

**Questions:**

Please see the weaknesses.

---

> ### Author Response · Authors · 2023-11-22
> **Response to comments by Reviewer cJRy**
>
> We thank the reviewer for their comments. Please see our responses below.
>
> **Q1: The novelty is limited. Formulating computer vision tasks as generative tasks is not novel and has been used in previous works[18]. The overall pipeline highly resembles InstructPix2Pix.**
>
> A1: While it is true that the idea of creating vision generalists has been explored before, this does not mean that our method is not novel. The paper in [18] cited by the reviewer uses visual prompting for in-context learning and does not involve any natural language component. While it is true that the architecture of our model resembles that of InstructPix2Pix due to the shared Pix2Pix formulation, the goal of our model, the way its training data is generated and the evaluation are completely different.
>
> **Q2: Performance is not impressive at all.**
>
> A2: The results in Section 4 indicate that the performance of our model surpasses the state-of-the-art vision generalists [13, 15, 17, 18, 19]. For tasks such as depth estimation and segmentation, our model outperforms other vision generalists on external datasets (Table 1) as well as in open dictionary setups (Table 2). We have also added a new comparison with the Painter model in [18] as well as an additional comparison with a task specific depth estimation model (See Tables below), both showing a competitive performance by our model. Note that our goal is not to outperform specialized models optimized on the target datasets, but rather to show the feasibility of instruction-guided execution of vision tasks and compare its performance and generalization abilities with comparable vision generalists.
>
> | Model         | Dep. Est. (RMSE↓)  |  Sem. Seg. (mIOU↑)| Cls. (Acc↑) | Det. (mAP\at0.5↑) |
> |:---------------:|:------------------:|:-----------------------:|:----------------:|:------------------:|
> |   Painter  | **0.288** (NYUv2) 0.285 (SUNRGB-D)  | 49.9 (ADE20k) 52.5 (VOC) |  - | -  |
> |   InstructCV-FP   | 0.275 (NYUv2) **0.268** (SUNRGB-D)  | **52.3** (ADE20k) **53.5** (VOC) |  80.4 (Pets) 75.1 (Imagenet-Sub) | 49.1(COCO) 62.0 (VOC) |
> |   InstructCV-RP  | 0.297 (NYUv2) 0.279 (SUNRGB-D)  | 47.2 (ADE20k) 52.1 (VOC) |  82.1 (Pets) 74.7 (Imagenet-Sub) | 48.5(COCO) 61.7 (VOC) |
>
> | Model         | Dep. Est. (RMSE↓) ｜
> |:---------------:|:------------------:|
> |   DINOv2 (SOTA)  | 0.279 (NYUv2)  |
> |   InstructCV-FP   | **0.275** (NYUv2)  |

---

### Official Review · Reviewer_QhMY · 2023-10-29

**Soundness:** 3 good
**Presentation:** 4 excellent
**Contribution:** 3 good
**Rating:** 8
**Confidence:** 3

**Summary:**

In this paper, the authors focus on the problem of a generalist visual model. To this end, they train a text-to-diffusion model on multiple tasks: Object Detection, Image Classification, Semantic Segmentation, and Depth Estimation. They use standard template phrases for each of these tasks and leverage a LLM to generate paraphrased variations of these questions and commands. They compare their visual model against other generalist models as well as task-specific models on each task. They show a level of performance which is competitive on most tasks.

**Strengths:**

1. The paper is very well organized and presented. Figures are very polished and simple to understand.

2. Given the strong performance of task-specific models and LLMs in recent years, the problem in question-generalist visual models-is a timely and relevant topic.

3. Quantitative performance (except for Image Classification) is impressive.

**Weaknesses:**

1. Performance on image classification is weak versus baselines. It is unclear if diffusion models are well suited to object classification versus the other tasks which are spatially grounded.

2. While generalist visual models are the frontier in Computer Vision, there should be more justification why such a system should rely on a single backend visual model versus a ensemble of experts (i.e. prompts to use Mask-RCNN for segmentation, ResNet for classification, etc.)

**Questions:**

1. Please add more justification in the paper specifically for the use of a single diffusion model versus an ensemble of experts.

2. Minor suggestion: Moving Section 3, Related Work, to come right after the introduction might help the paper flow better.

---

> ### Author Response · Authors · 2023-11-22
> **Response to comments by Reviewer QhMY**
>
> We thank the reviewer for the thoughtful comments and feedback.
>
> **Q1: The necessity of using a single diffusion model compared to an ensemble of experts.**
>
> A1: An ensemble of experts, where Mask-RCNN is used for segmentation, ResNet for classification, etc., necessitate training each specialized component separately on a predefined set of categories. This type of model lacks the ability to generalize to new categories not covered in the training data. In contrast, our approach employs a single unified model, eliminating the need for training separate models for each task. This unified model can effectively transfer knowledge across various tasks and categories.
>
> **Q2: Suggestions about the structure of the paper.**
>
> A2: Thank you for the suggestion, we will make the necessary modifications in the final version of the paper.

---

### Official Review · Reviewer_ERkX · 2023-10-30

**Soundness:** 3 good
**Presentation:** 3 good
**Contribution:** 3 good
**Rating:** 6
**Confidence:** 3

**Summary:**

This paper introduces InstructCV, a unified language interface for computer vision tasks that enables task execution through natural language instructions. It leverages text-to-image generation by casting various computer vision tasks as instruction-based image synthesis problems. By creating a multi-modal and multi-task training dataset and applying instruction-tuning to a text-to-image diffusion model, InstructCV achieves competitive performance and compelling generalization capabilities compared to other vision models.

**Strengths:**

1. A valuable contribution of the paper is the introduction of a novel approach to integrate dense prediction tasks with a T2I model. Specifically, the paper suggests utilizing a generative model to perform a task similar to editing, which seems reasonable for tasks like segmentation, and depth prediction compared to sequence to sequence model.
2. With the help of T2I models, the algorithm has demonstrated good generalization ability in dealing with unseen categories.

**Weaknesses:**

1. The unique advantage of instruction tuning lies in its ability to generalize to unseen tasks[1], thereby enabling the model to become a general-purpose model. In the article, I believe it resembles more of a multi-task learning approach since the instructions are mapped to several tasks mentioned in the article. In my opinion, this approach does not have clear advantages compared to simply performing these tasks in a multitask learning setting on T2I model.
2. For certain tasks that do not require dense output, such as classification and detection, it may not be necessary to forcefully address them as image generation tasks. Sequence models are generally more suitable for modeling such tasks, and the approach described in the article seems highly inappropriate in such cases.

[1] InstructBLIP: Towards General-purpose Vision-Language Models with Instruction Tuning

**Questions:**

1. The advantage of instruction tuning lies in its ability to enable the model to generalize to unseen tasks, which sets it apart from simple multi-task learning. However, in the paper titled "InstructCV," it seems that the approach primarily identifies instructions as task indicators within a predefined set of tasks. This may not demonstrate an advantage over traditional multitask learning. Could the author provide further explanation regarding the unique advantages of this approach?
2. I believe the good generation ability to new datasets is mostly because the comparison baselines lack the ability to handle open vocabulary. On the other hand, T2I  models are trained on a large amount of text-to-image pairs, which naturally gives them a good open vocabulary capability. Additionally, some unified models like x-decoder also demonstrate excellent generalization ability to new categories.

In summary, one aspect that concerns me is the lack of clear advantages in unifying dense prediction as an image-to-image generation task. The article mentions instruction tuning, but the experimental results do not generalize to new tasks. The open vocabulary capability is primarily attributed to the T2I models, and I don't believe this alone can be considered unique because The open vocabulary capability has also demonstrated impressive performance in previous works, such as xdecoder[1].

[1] X-Decoder: Generalized Decoding for Pixel, Image and Language

---

> ### Author Response · Authors · 2023-11-22
> **Response to comments by Reviewer ERkX**
>
> We thank the reviewer for the thoughtful comments and feedback. Please see below a point-by-point response.
>
> **Q1: Could the author provide further explanation regarding the unique advantages of this approach?**
>
> A1: Thank you for this thought-provoking comment. We realize that the way we implement instruction-tuning in our setup may seem different from its application in NLP, where the primary goal is zero-shot generalization to entirely unseen tasks. However, we attribute this distinction to the structural differences between language and vision tasks. Vision tasks inherently involve more structure, where the anatomy of a vision task combines a canonical task type (e.g., segmentation) with a specific category or object. **In this narrower sense, we believe that InstructCV exhibits generalization to new tasks by effectively handling segmentation, classification, and detection for categories not encountered during the instruction-tuning dataset training.** This capability arises from the T2I formulation of InstructCV, allowing it to comprehend both the intended tasks and the associated categories and apply the learned task to new categories in an open dictionary fashion. **We consider this a unique advantage of our approach**—the ability to leverage semantic correspondence between instructions and images to establish a shared representation capable of generalizing the four tasks to new categories within an open dictionary framework.
> Furthermore, we believe that our model takes initial steps toward zero-shot generalization to entirely new tasks beyond the four considered. The rephrased instruction training equips the model to understand language descriptions of tasks, which fundamentally differs from the rigid task indicators of traditional multi-task learning— this flexibility is another key advantage of our model. For instance, one could envision implementing a panoptic segmentation task in a zero-shot manner by providing a natural language description that combines semantic segmentation and classification tasks. Although such capabilities may fully emerge in a future version of InstructCV trained on a larger number of tasks, the rephrased model represents a foundational step towards achieving this goal.
>
> We will incorporate the discussion above in the final version of the paper.
>
> **Q2: I believe the good generalization ability to new datasets is mostly because the comparison baselines lack the ability to handle open vocabulary. On the other hand, T2I models are trained on a large amount of text-to-image pairs, which naturally gives them a good open vocabulary capability. Additionally, some unified models like x-decoder also demonstrate excellent generalization ability to new categories.**
>
> A2: Yes, the strong generalization ability of T2I which is trained on a very large dataset and that is the reason why we choose T2I to achieve the Vision Generalists. This is the key idea behind InstructCV; the pretrained multimodal representation can be repurposed to create a model that implements vision tasks in open vocabulary generalization.
> Thank you for bringing up the paper on X-decoder. While it is indeed related to our work, the authors in that paper have slightly different motivations and goals. We appreciate this insight and will incorporate a discussion on it in the final version of our paper.

---

### Official Review · Reviewer_wRB5 · 2023-10-31

**Soundness:** 2 fair
**Presentation:** 2 fair
**Contribution:** 2 fair
**Rating:** 5
**Confidence:** 5

**Summary:**

This paper proposes a vision generalist, InstructCV, which casts various computer vision tasks as text-guided image generation. Based on a text-guided image editing model, InstructCV learns to generate visually encoded outputs of different vision tasks. Experimental results on several visual recognition tasks showcase the effectiveness of InstructCV.

**Strengths:**

1. Different from previous attempts to build vision generalists, this paper provides a solution built upon text-to-image diffusion models. Task prompting is achieved by text instructions, which are human-intuitive and general.

2. The experimental results indicate the validity of InstructCV in handling a variety of visual recognition tasks.

**Weaknesses:**

1. After reading the paper, I am a bit confused about the visually encoded outputs for different tasks.
a) Semantic segmentation: How to derive the mask of all semantic classes for an input image? If predicting them one by one and each class is manually indicated, I would say it is more like a task of referring segmentation rather than semantic segmentation. It would be helpful if the authors could provide more elaboration on this.
b) Object detection: What do the authors mean by saying “cross-reference the bounding boxes with the dataset annotations” in Appendix A.3?

2. What type of instructions is used during inference? Fixed templates or random rephrased ones? Does it remain the same for InstructCV-FP and InstructCV-RP during inference? As reported in Table 3, InstructCV-FP outperforms InstructCV-RP if both use template prompts during inference, so why bother training InstructCV using rephrased prompts? I understand this could improve the robustness of variations in wording, but one can just adopt a fixed template prompt to address a standard computer vision task. In addition, the rephrased prompts for each task are not that diverse (only 7 variations according to Table 4), why do we need to leverage LLMs as this could be readily done by humans? I would like to see more discussions about the benefits brought by utilizing rephrased prompts.

3. More ablations should be conducted to compare InstructCV-FP with InstructCV-RP. For example, the authors could report the results of InstructCV-FP in Table 1.

4. For the organization of this paper, I suggest the authors move the introduction of diffusion models in Section 2.2 (which takes up most of this section) to a “Prerequisites” subsection or Appendix.

5. The proposed method claims to be a vision generalist but it is validated only in some visual recognition tasks. What about other vision tasks such as low-level tasks (e.g., denoise, deblur, and derain) and generation (as done in Prompt Diffusion [1*])?

6. Further comparisons with other SOTA unified models such as Painter [2*] should be conducted. Painter is currently performing the best in prediction tasks as a vision generalist.

7. In terms of the computational cost, the authors claim that InstructCV exhibits much faster inference speed than previous methods such as Unified-IO. However, InstructCV needs multiple inferences for some tasks (e.g., semantic segmentation and classification), which could consume N×times where N is the number of semantic classes. Besides, the reported inference time of InstructCV on a single NVIDIA A100 GPU for an image is 5 seconds in the main paper but 10 seconds in the Appendix.

8. In the second paragraph of Section 1, the authors claim that “However, these prompts consist of (uninterpretable) numerical values derived from specific training datasets, which may limit their ability to generalize to new datasets, tasks, or categories”. Why do the prompts used in inpainting-based methods (which are visual examples) contain uninterpretable numerical values?

9. The format of references is messy and inconsistent. For example, a bunch of references do not list the conference/journal title.

10. Just a reminder that there is a very similar work, InstructDiffusion [3*], to this submission. The authors could consider citing that paper and provide comparisons and discussions.

[1*] In-context learning unlocked for diffusion models. arXiv, 2023.

[2*] Images speak in images: A generalist painter for in-context visual learning. In CVPR, 2023.

[3*] InstructDiffusion: A Generalist Modeling Interface for Vision Tasks. arXiv, 2023.

**Questions:**

See weaknesses.

---

> ### Author Response · Authors · 2023-11-22
> **Response to comments by Reviewer wRB5**
>
> We thank the reviewer for the thoughtful comments. Please see below a point-by-point response.
>
> **Q1: Clarification of the visual encoding of outputs for different tasks.**
>
> **Q1.1:** *Distinction between semantic segmentation and referring segmentation*
>
> A1.1: We appreciate the reviewer's comment. It is accurate to note that, like other baselines such as "Unified-IO" and "Inpainter," our approach to encoding and implementing segmentation is analogous to referring image segmentation. Note that this is a consequence of our framing of *all* vision tasks within a common text-to-image generation setup, which means that all segmentation tasks are prompted by natural language instruction. Semantic segmentation can be implemented by querying classes one-by-one. We focused on the more standard semantic segmentation task in this paper since many of the baselines (e.g. the specialized architectures) cannot condition segmentation on natural language inputs. We will add a discussion on this issue in the final version of the paper to clear up any confusion.
>
> **Q1.2:** *What do the authors mean by "cross-reference the bounding boxes with the dataset annotations" in Appendix A.3?*
>
> A1.2: The post-processing steps for the object detection task involve obtaining the coordinates of the bounding boxes displayed in the visual output of InstructCV using out-of-the-box post-processing functions in the OpenCV library. After that, we calculate the mAP@0.5 metric using the standard procedure in the COCOeval library. The “cross-referencing” statement was meant to describe the evaluation and not the post-processing procedure, but was misplaced in the Appendix. We will fix this error in the final version of the paper.

---

> ### Author Response · Authors · 2023-11-22
> **Response to comments by Reviewer wRB5 (Cont'd)**
>
> **Q2: Utility of training using rephrased instructions**
>
> **Q2.1:** *What type of instructions is used during inference?*
>
> A2.1: In Tables 1 and 2, we use randomly generated rephrased prompts during inference for evaluating the InstructCV-RP model. These rephrased prompts were sampled at inference time and were not drawn from the prompts used for training.
>
> **Q2.2:** *Why training InstructCV using rephrased prompts and why do we need to leverage LLMs as this could be readily done by humans?*
>
> A2.2: Please note that the InstructCV-RP model was not trained only on the 7 prompts in Table 4. The 7 rephrased prompts in this table are illustrative examples of the sampled prompts from the LLM (as highlighted in the Table caption). To achieve good generalization performance, the model was trained on a large number of rephrased prompts repeatedly sampled from the LLM. We will emphasize this in the final version of the paper.
> The usage of the LLM is useful for both creating a scalable framework for training InstructCV on diverse rephrased prompts, as well as evaluating the model on truly diverse and seen rephrasing of the original instructions by repeatedly sampling new prompts at inference time. To further illustrate the generalization performance of InstructCV-RP to new instructions, we conducted further experiments of the segmentation performance in the ADE20k dataset as highlighted below.
>
> |              Instructions               | InstructCV-FP (mIOU Score) | InstructCV-RP (mIOU Score) |
> |---------------------------------------|:-------------------------:|:-------------------------:|
> | Template prompt: Segment the %category%. |   45.503    | 42.793 |
> | In what parts of the image can the %category% be found? | 21.344 | 45.600 |
> | Can you please segment the %category%? | 40.300 | 45.810 |
> | Hi, InstructCV, segment the %category%. | 39.980 | 46.450 |
> | Which portions of the image encompass the %category%?| 15.223 | 42.345 |
> | Could you point out the areas in the image that have the %category%? | 25.400 | 41.560 |
> | What regions of the image feature the %category%? | 19.523 | 45.600 |
> | I would appreciate it if you could help segment the %category%.  | 36.712 | 45.800 |
> | I would be grateful if you could assist in segmenting the %category%. | 26.764 | 43.260 |
>
> Additionally, we tested the robustness of InstructCV-RP to prompt rephrasing through a different LLM (GPT-4), and calculated the average and variance of the results. The findings highlighted below indicate the robustness of InstructCV-RP to unseen instructions.
>
> |              Our model               | Mean (mIOU Score)  | Variance (mIOU Score) |
> |---------------------------------------|:-------------------------:|:-------------------------:|
> | InstructCV-FP. |   25.618    | 127.850 |
> | InstructCV-RP. |   45.503    | 5.85 |
>
> **Regarding the utility of rephrased prompts:** We would like to stress that the primary goal of this paper is to create a model capable of performing standard computer vision tasks based on natural language instructions. This approach aims to enhance the model's ability to generalize to both **(1)** previously unseen categories and **(2)** novel user-generated instructions.
>
> The importance of generalizing to unseen categories lies in the model's capacity to apply knowledge gained from training on specific categories in one dataset to perform tasks on entirely new categories in a different dataset. In the context of InstructCV, its ability to achieve this (**Goal 1**) is attributed to its text-to-image formulation, a quality applicable to both fixed and rephrased prompts.
>
> While it is true that the same vision tasks can be executed using fixed prompts, this is not unique to InstructCV and is true for any multi-task model not reliant on natural language inputs. The utilization of rephrased prompts serves a different purpose: it does not aim to enhance performance on standard tasks but rather to improve the model's adaptability to diverse instructions and expressions from end users (**Goal 2**).
>
> These capabilities are envisioned to be particularly beneficial for achieving zero-shot generalization to entirely new tasks beyond the four fixed tasks used in training. For instance, one could envision implementing a panoptic segmentation task in a zero-shot manner by providing a natural language description, combining semantic segmentation and classification tasks. While such capabilities are only likely to emerge in a version of InstructCV that is trained on a larger number of tasks, the rephrased model represents an initial step towards this goal.

---

> ### Author Response · Authors · 2023-11-22
> **Response to comments by Reviewer wRB5 (Cont'd)**
>
> **Q3: More ablations should be conducted to compare InstructCV-FP with InstructCV-RP. For example, the authors can report the results of InstructCV-FP in Table 1.**
>
> A3: We have added the performance of InstructCV-FP in Table 1.
>
> | Model         | Dep. Est. (RMSE↓)  |  Sem. Seg. (mIOU↑)| Cls. (Acc↑) | Det. (mAP\at0.5↑) |
> |:---------------:|:------------------:|:-----------------------:|:----------------:|:------------------:|
> |   InstructCV-FP   | 0.275 (NYUv2) 0.268 (SUNRGB-D)  | 52.3 (ADE20k) 53.5 (VOC) |  80.4 (Pets) 75.1 (Imagenet-Sub) | 49.1(COCO) 62.0 (VOC) |
> |   InstructCV-RP  | 0.297 (NYUv2) 0.279 (SUNRGB-D)  | 47.2 (ADE20k) 52.1 (VOC) |  82.1 (Pets) 74.7 (Imagenet-Sub) | 48.5(COCO) 61.7 (VOC) |
>
> We will also include both versions of InstructCV to all the performance comparison Tables.
>
> **Q4: For the organization of this paper, I suggest the authors move the introduction of diffusion models in Section 2.2 (which takes up most of this section) to a “Prerequisites” subsection or Appendix.**
>
> A4: Thanks for the suggestion. We will make this change in the final version of the paper.
>
> **Q5: The proposed method claims to be a vision generalist but it is validated only in some visual recognition tasks. What about other vision tasks such as low-level tasks (e.g., denoise, deblur, and derain) and generation (as done in Prompt Diffusion)?**
>
> A5: The InstructCV pipeline is not limited to the 4 tasks considered in the paper, but can be applied to a much broader range of vision tasks including low-level tasks as well. Following [18], we fine-tuned InstructCV to conduct 3 more low-level vision tasks: Denoising (SIDD dataset), Deraining (Rain14000, Rain800, Rain100H, Rain100L, Rain1200 datasets) and Enhance (LOw-Light dataset). (Due to the time constraints, we were only able to evaluate the InstructCV-FP model.) Results are provided below.
>
> |               | Denoising (SIDD) | Deraining (5 datasets) | Enhance |
> | --------------- |:---------------:|:---------------:|:---------------:|
> | Painter  | 38.88 (PSNR$\uparrow$)  **0.954** (SSIM$\uparrow$)  | 29.49 (PSNR$\uparrow$)  0.868 (SSIM$\uparrow$) | 22.40 (PSNR$\uparrow$)  0.872 (SSIM$\uparrow$)  |
> | InstructCV-FP   | **39.20** (PSNR$\uparrow$)  0.949 (SSIM$\uparrow$)  | **31.47** (PSNR$\uparrow$)  **0.872** (SSIM$\uparrow$)  | **24.82** (PSNR$\uparrow$)  **0.895** (SSIM$\uparrow$) |
>
> We will add the results on low-level tasks in the final version of the paper.
>
>
> **Q6: Further comparisons with other SOTA unified models such as Painter should be conducted. Painter is currently performing the best in prediction tasks as a vision generalist.**
>
> A6: Thanks for this suggestion. We have added Painter as a baseline in Table 1 (See below). On tasks supported by Painter, our InstructCV-FP model achieved higher performance on external datasets, and on non-external datasets, InstructCV-FP also achieved higher performance except for NYUv2.
>
> | Model         | Dep. Est. (RMSE↓)  |  Sem. Seg. (mIOU↑)| Cls. (Acc↑) | Det. (mAP\at0.5↑) |
> |:---------------:|:------------------:|:-----------------------:|:----------------:|:------------------:|
> |   Painter  | **0.288** (NYUv2) 0.285 (SUNRGB-D)  | 49.9 (ADE20k) 52.5 (VOC) |  - | -  |
> |   InstructCV-FP   | 0.275 (NYUv2) **0.268** (SUNRGB-D)  | **52.3** (ADE20k) **53.5** (VOC) |  80.4 (Pets) 75.1 (Imagenet-Sub) | 49.1(COCO) 62.0 (VOC) |
> |   InstructCV-RP  | 0.297 (NYUv2) 0.279 (SUNRGB-D)  | 47.2 (ADE20k) 52.1 (VOC) |  82.1 (Pets) 74.7 (Imagenet-Sub) | 48.5(COCO) 61.7 (VOC) |
>
> **Q7: Comparing inference speeds with baselines.**
>
> A7: We believe that our claim on InstructCV exhibiting much faster inference speed than previous methods such as Unified-IO is fair, because Unified-IO also infers one class at a time, and the inference times we correspond to speed of inference per class. The inference time is 5 seconds and not 10 seconds, the discrepancy arises from an earlier version of our model. We will fix this error in the Appendix of the final paper.

---

> ### Author Response · Authors · 2023-11-22
> **Response to comments by Reviewer wRB5 (Cont'd)**
>
> **Q8: In the second paragraph of Section 1, the authors claim that “However, these prompts consist of (uninterpretable) numerical values derived from specific training datasets, which may limit their ability to generalize to new datasets, tasks, or categories”. Why do the prompts used in inpainting-based methods (which are visual examples) contain uninterpretable numerical values?**
>
> A8: We appreciate the reviewer for bringing this to our attention. The statement in question was intended to address **prompt tuning** approaches. These approaches involve learning task-specific "codes," such as input tokens or pixels, to guide Seq2Seq models in executing the desired task. This specifically pertains to the first category of methods discussed in paragraph 2 on page 2. It's important to note that this doesn't apply to [18] and [19], as they learn the vision task in-context using interpretable visual examples. To avoid any confusion, we will explicitly state that this statement is referring to **prompt tuning** and not **in-context** approaches.
>
> **Q9: The format of references is messy and inconsistent. For example, a bunch of references do not list the conference/journal title.**
>
> A9:  Thanks for pointing this out. We will revise the formatting of references in the final version of the paper.
>
> **Q10: Just a reminder that there is a very similar work, InstructDiffusion, to this submission. The authors could consider citing that paper and provide comparisons and discussions.**
>
> A10: Thank you for your suggestions. We were not aware of InstructDiffusion at the time of submission, as it is concurrent with our work. We appreciate your diligence in bringing this to our attention. To address this, we will include a citation for InstructDiffusion and incorporate discussions about it in the paper.

---

> > ### Comment · Reviewer_wRB5 · 2023-11-22
> > **remaining questions**
> >
> > Thanks for the detailed response. Some remaining concerns are as follows:
> >
> > For Q1.1: If semantic segmentation is fulfilled by querying classes one by one, how does one know the set of classes in advance and how can one decide whether the query at a time is a success (especially in the case without manual intervention)? I still think leveraging InstructCV to perform semantic segmentation and classification tasks is a bit awkward. I would suggest directly assessing it by referring to segmentation.
> >
> > For Q2.2: I understand that one benefit brought by utilizing rephrased prompts is the better generalization to new instructions given by users during inference. However, this is not appealing given that InstructCV-FP attains better performance than InstructCV-RP (see Table 3 in the main paper and the table provided in A3). The authors further claimed that adopting rephrased prompts is beneficial for the generalization to previously unseen categories, but I found this is not the case. As shown in the table provided in A3, InstructCV-FP still outperforms InstructCV-RP on external datasets.
> >
> > Further, I do not see a revised paper to reflect the changes and additional experiments.

---

> ### Author Response · Authors · 2023-11-23
> **Incorporation of modifications and new results in the revised paper**
>
> Thank you so much for the prompt feedback. We apologize for not reflecting the changes and new experiments in a revised manuscript, we did not realize that we can upload a revision during the rebuttal period. We did our best to incorporate as many of the changes mentioned above as well as the new experiments in the revised version. In particular, we incorporated the following changes:
>
> **Q1.2:** We clarified the statement on "cross-referencing the bounding boxes with the dataset annotations" in Appendix A.3.
>
> **Q2.1:** We clarified the types of instructions used during inference with InstructCV.
>
> **A2.2:** We clarified that the InstructCV-RP model was not trained only on the 7 prompts in Table 4.
>
> **Q3:** We incorporated the results of InstructCV-FP in Table 1.
>
> **Q5:** We added the results on. low level vision tasks in a new Table in the Appendix.
>
> **Q6:** We added the results of the Painter baseline in [18] to Table 1.
>
> **Q7:** We clarified our comments about the inference speed and its comparison with that of Unified-IO.
>
> **Q8:** We clarified our comment on the difference between prompt tuning and in-context learning.
>
> **A10:** We added a citation to InstructDiffusion.
>
> We will incorporate all the remaining changes in the final paper.

---

> ### Author Response · Authors · 2023-11-23
> **Response to remaining questions (I)**
>
> **Q1.1: On using referring segmentation instead of semantic segmentation.**
>
> A1.1: Thank you for your suggestion. We agree that referring segmentation (and not semantic segmentation) would be a more natural task to incorporate in our pipeline, since as the reviewer points out, sequential segmentation of classes would require a manual prespecification of these classes. The reason we initially considered semantic segmentation is that this would enable us to compare performance with existing vision generalists which do not incorporate a natural language component.
> We conducted a new evaluation experiment focused on referring segmentation, and will replace semantic segmentation with this task in the final manuscript. We followed the work in [C1] in preparing the datasets, and used overall mean Intersection over Union (oIoU) as an evaluation metric. We considered three datasets RefCOCO val, RefCOCO test A, RefCOCO test B for evaluation. We did not evaluate other vision generalists (such as Painter and Inpainting), because they use visual prompts for semantic segmentation, and are not naturally fit for conducting referring segmentation.
> Please note that **InstructCV does not use the RefCOCO dataset to train the segmentation task**, but as the results below show, it can achieve **competitive performance** compared to **specialized model training on the RefCOCO dataset**. Further, InstructCV significantly outperformed the Unified-IO vision generalist.
>
>
> | Model         | RefCOCO val (oIoU)  |  RefCOCO test A (oIoU)  | RefCOCO test B (oIoU)  |
> |:---------------:|:------------------:|:-----------------------:|:----------------:|
> |   Specialized model: DMN [C2]  |     49.78        |       54.83        |        45.13          |
> |   Specialized model: BRINet [C3]  |     60.98     |      62.99   |   	59.21  	   |
> |   Vision generalist: Unified-IO  |       28.80       |        32.58        |        25.32        |
> |   Vision generalist: InstructCV-FP   |     **53.20**     |     **54.98**     |       **51.45**  |

---

> ### Author Response · Authors · 2023-11-23
> **Response to remaining questions (II)**
>
> **Q2: I understand that one benefit brought by utilizing rephrased prompts is the better generalization to new instructions given by users during inference. However, this is not appealing given that InstructCV-FP attains better performance than InstructCV-RP (see Table 3 in the main paper and the table provided in A3). The authors further claimed that adopting rephrased prompts is beneficial for the generalization to previously unseen categories, but I found this is not the case. As shown in the table provided in A3, InstructCV-FP still outperforms InstructCV-RP on external datasets.**
>
> A2: We believe that Table 3 shows that InstructCV-RP outperforms InstructCV-FP on almost all rephrased instructions. In the new result in A3, InstructCV-FP was evaluated on fixed prompt whereas InstructCV-RP was evaluated on rephrased prompts since we believe that we would be a more fair comparison (i.e., InstructCV-FP was not designed to handle varying user-written instructions). Thus, the differences in performance result from differences in the evaluation procedure and not the capabilities of the two variants of our model. If both models are to be evaluated with the same prompt template, InstructCV-RP outperforms InstructCV-FP on most tasks as shown in the Table below.
>
> | | Dep. Est. (RMSE↓) | Ref. Seg. (oIOU↑) | Cls. (Acc↑) | Det. (mAP@0.5↑)  |
> |---------------|------------------|-----------------------|----------------|----------------|
> |  | Estimation the depth of this image | Segment the “%Category” | Show “%Color” if there has “%Category” | Detect the “%Category” |
> |   InstructCV-FP  |     0.275 (NYUv2) 0.268 (SUNRGB-D)      |  53.20 (RefCOCO val)    |  80.4 (Pets) 75.1 (Imagenet-Sub)        |      49.1(COCO) 62.0 (VOC)        |
> |   InstructCV-RP (tested with Fixed Prompts)  |     **0.270** (NYUv2) **0.260** (SUNRGB-D)    |   **54.90** (RefCOCO val)    |  78.3 (Pets) 73.2 (Imagenet-Sub)   |   **50.5** (COCO) **63.2** (VOC)	  |
>
> We also believe that the reviewer may have misinterpreted our response regarding the usefulness of rephrased prompts. In our previous response **A2.2**, we mentioned that our model has two goals of generalizing to both (1) previously unseen categories and (2) novel user-generated instructions. As we mentioned in **A2.2**, the utilization of rephrased prompts does not aim to enhance performance on standard tasks but rather to improve the model's adaptability to diverse instructions and expressions from end users (Goal 2), which we believe is demonstrated in Table 3. We did not claim that rephrased prompts enable generalization to unseen categories (Goal 1), as mentioned in **A2.2**, these capabilities result from the text-to-image formulation of the vision tasks. Overall, we think the rephrased prompts do not hurt the performance and enable better generalization to different ways of writing the same instructions in natural language.
> Finally, we wanted to stress that the rephrased prompting is only a variant of our original model and is not the key message of our paper. While we have not tested the ability of our model to utilize its language comprehension to execute more complex tasks (panoptic segmentation or compositional tasks) in a zero-shot fashion, we believe InstructCV-RP takes first steps towards building models that can implement completely new vision tasks based purely on language descriptions.
>
> **References**
>
> [C1] Henghui Ding, Chang Liu, Suchen Wang, Xudong Jiang. Vision-Language Transformer and Query Generation for Referring Segmentation. ICCV 2021.
>
> [C2] Edgar Margffoy Tuay, Juan C Perez, Emilio Botero, and Pablo Arbelaez. Dynamic multimodal instance segmentation guided by natural language queries. ECCV 2018.
>
> [C3] Zhiwei Hu, Guang Feng, Jiayu Sun, Lihe Zhang, and Huchuan Lu. Bi-directional relationship inferring network for referring image segmentation. CVPR 2020.

---

### Meta-Review · Area_Chair_TLXZ · 2023-12-16

**Metareview:**

**Scientific Claims and Findings**:
The paper introduces InstructCV, a vision generalist model that leverages text-to-image diffusion models for various computer vision tasks. InstructCV is trained on a multi-modal and multi-task dataset, using text instructions to guide image generation for tasks like object detection, image classification, semantic segmentation, and depth estimation. The model demonstrates competitive performance across tasks, and the paper explores the effectiveness of instruction-tuning for generalization.

**Strengths of the Paper**:

- Innovative Approach: The paper introduces a novel approach to vision generalization by combining text-to-image diffusion models with natural language instructions for various tasks.
- Multi-Task Training: InstructCV is trained on a diverse set of tasks, showcasing its ability to handle different computer vision challenges through a unified model.
- Generalization Capability: The model exhibits good generalization to unseen categories and tasks, providing a valuable contribution to the development of general-purpose vision-language models.
- Clear Presentation: The paper is well-organized, with polished figures and a straightforward presentation, making it easy for readers to understand the proposed approach.

**Weaknesses of the Paper**:

- Classification Performance: InstructCV shows weaker performance in image classification compared to other tasks. The paper should provide more insights into the limitations or challenges specific to classification within the proposed framework.
- Ensemble vs. Single Model: The paper lacks sufficient justification for why a single diffusion model is chosen over an ensemble of task-specific models. Considering the diversity of computer vision tasks, discussing the trade-offs and advantages of a single model approach would enhance the paper.
- Task Specificity: For tasks like classification and detection that don't require dense output, the approach of addressing them as image generation tasks may not be optimal. The paper could discuss the suitability of the proposed method for various types of tasks.
- Comparisons with Other Models: The paper could benefit from further comparisons with state-of-the-art unified models, such as Painter, to provide a more comprehensive assessment of InstructCV's performance.
- Justification for Instruction Tuning: While the paper mentions the advantage of instruction tuning for generalization, more detailed explanations about the unique benefits of this approach compared to traditional multi-task learning would enhance clarity.

**Justification For Why Not Higher Score:**

- More Ensemble Discussion: The paper lacks detailed discussions on the potential benefits and drawbacks of using an ensemble of task-specific models compared to a single diffusion model.
- Clearer Justification: The paper needs a more detailed justification for the choice of a single diffusion model and its implications for handling diverse computer vision tasks.

**Justification For Why Not Lower Score:**

The idea of leveraging diffusion models for various CV recognition tasks in an instruction-following setting is new.

---

### Decision · Program_Chairs · 2024-01-16

Accept (poster)